# The Effect of *Microcystis* on the Monitoring of Faecal Indicator Bacteria

**DOI:** 10.3390/toxins15110628

**Published:** 2023-10-26

**Authors:** Jingjing Zhou, Mingzhi Qu, Christy Dunkinson, Daniel D. Lefebvre, Yuxiang Wang, R. Stephen Brown

**Affiliations:** 1School of Environmental Studies, Queen’s University, Kingston, ON K7L 3N6, Canada; jingjing.zhou@queensu.ca (J.Z.); cedunk2222@gmail.com (C.D.); 2Department of Biology, Queen’s University, Kingston, ON K7L 3N6, Canada; qumingzhi1986@gmail.com (M.Q.); yuxiangw@queensu.ca (Y.W.); 3Department of Chemistry, Queen’s University, Kingston, ON K7L 3N6, Canada

**Keywords:** cyanobacteria, *Escherichia coli*, *Microcystis*, microcystin, faecal indicator bacteria

## Abstract

The survival of *Escherichia coli* (*E. coli*) bacteria, the most common faecal indicator bacteria (FIB), may be significantly affected by cyanobacteria present during a harmful algal bloom (HAB). Therefore, the effect of *Microcystis* on the survival of FIB *E.coli* and coliforms was investigated. Microcosms containing two species of *Microcystis* (*M. aeruginosa* and *M. smithii*) were established and then inoculated with four reference strains of *E. coli* (ATCC 25922, 8739, 51813, and 11775) to explore the cyanobacteria-bacteria dynamics at a laboratory setting. Monitoring over several days showed normal growth of *Microcystis*, with or without the presence of *E. coli*. However, *Microcystis* was shown to dramatically decrease the survival of *E. coli* over time. Analysis of microcystin production by *Microcystis* was found to correlate with loss of *E. coli*, suggesting a toxic effect of microcystins on *E. coli* bacteria. This phenomenon was also demonstrated for a natural consortium of *E. coli* and coliform bacteria by inoculating with contaminated lake water. The results indicate that the use of *E. coli* as FIB may be greatly compromised in the presence of *Microcystis* spp. such as during a HAB when associated toxins are produced.

## 1. Introduction

Water bodies in North America and around the world have experienced increased algal blooms, including harmful algal blooms (HABs) caused by cyanobacteria as a result of enhanced human- and climate-driven environmental changes [1]. HABs affect the ecological integrity of the affected water by disrupting or altering the biological communities. Cyanotoxins such as microcystins that are produced by cyanobacteria, most notably *Microcystis* species, are known to cause numerous toxic effects in mammals, other animals and plants [2]. Although these compounds can negatively impact the growth and carbon fixation of photosynthetic prokaryotes [3], there is little information concerning their effects on heterotrophic prokaryotes [4,5,6]. Even though changes in bacterial assemblages have been detected in response to *Microcystis* colonies [7] and microcystins [8], there are no available reports addressing directly the dynamics of cyanobacteria and *E. coli* or other coliform bacteria, especially from the point of view of the effectiveness of faecal indicator bacteria (FIB) to indicate water quality when there is a risk of HABs being present. Harmful impacts on species such as FIB could significantly impact water management monitoring and any consequential remediative practices.

Transmission of microbial pathogens by water is a significant cause of waterborne illnesses worldwide [9,10]. It is impractical to perform tests for all potential pathogenic microbes in water. Furthermore, many waterborne pathogens tend to travel through the environment and may be diluted below normal detection limits, even if still present at clinically effective concentrations. These limitations led to the birth of the ‘indicator’ approach: to use easily detectable and representative microorganisms that are normally present in faecal pollution to indicate the possible presence of faecal pathogens, and therefore, an elevated risk of microbiological contamination in the water [11,12]. An effective indicator organism must fulfill certain criteria, such as having similar sources and fate as the associated pathogens, and there must be readily available, easy and affordable tests for the indicator [11]. Since the majority of pathogenic microbes are derived faecally, FIB are utilized widely to assess microbial water quality, including *E. coli*, total coliforms, faecal coliforms, faecal streptococci, and *Enterococci* [13]. *E. coli* bacteria have emerged as the preferred FIB as a result of demonstrated correlation with pathogen presence [11] along with the availability of a number of reliable, low-cost testing methods [14,15].

While significant attention has been paid to the correlation between FIB and pathogen sources, less attention has been devoted to assessing whether FIB and pathogens have the same fate in environmental waters. To interpret better the FIB absence/presence/abundance as a surrogate for microbial water quality, it is crucial to understand the dynamics and association of FIB and coexistent microorganisms in natural waters.

This paper reports an investigation into the growth pattern of selected FIB in the presence of the cyanobacteria *Microcystis* in laboratory settings. *Microcystis* was chosen because this genus is a major contributor to cyanobacterial harmful algal blooms, and the cytotoxin microcystin, in Canada and the United States [16,17,18]. Microcosms containing FIB (*E. coli* consortium) and cyanobacteria (*Microcystis aeruginosa*; *Microcystis smithii*) were prepared, and the water was sampled regularly to monitor the growth of both FIB and cyanobacteria. *E. coli* was severely suppressed by the presence of both *Microcystis* species, especially *M. aeruginosa*. Possible contributing factors include microcystins and cellular contact.

## 2. Results

### 2.1. Survival of E. coli Consortium in Microcystis Cultures

*M. aeruginosa* and *M. smithii* were cultured for 3 days with MA medium [19] and survival of *E. coli* inoculated at 40,000 CFUs/100 mL in the MA medium with and without *Microcystis* is shown in Figure 1. CFU refers to “colony forming units”, which is the standard unit of microbiological culture methods used to enumerate bacteria, but can be compared with cells in this context. For comparison with *Microcystis* levels (see below), the *E. coli* inoculation was at 0.4 cells/µL.

*E. coli* amounts are normally stable in water bodies after leaving their enteric habitats, but the *E. coli* level increased slightly in MA medium, which contains a carbon source in the form of 0.163 mM β-glycerophosphate, and then its level slowly decreased but remained above the initial level for at least 28 days. When inoculated into the *M. smithii* culture, the *E. coli* level dropped quickly, with no *E. coli* being detected after 8 days. When inoculated with *M. aeruginosa*, the *E. coli* counts dropped more rapidly, with no *E. coli* being detected after 2 days. 

The decrease in *E. coli* counts in Figure 1 was analyzed assuming a pseudo-first-order decay (see Materials and Methods) by regression analysis of log_e_-transformed *E. coli* counts versus days from inoculation. Half-life values (t12), defined as the time required for decay to reduce the population by 50%, were determined along with 95% confidence intervals (95% CI) reported as the 95% CI range of half-life values. The *E. coli* inoculated into *M. smithii* culture decreased with a half-life of 0.67 days (95% CI 0.59 to 0.79 days), while the *E. coli* inoculated into *M. aeruginosa* culture decreased with a half-life of 0.21 days (95% CI 0.14 to 0.47 days), consistent with the observed times for loss of bacteria. The control culture, starting from day 1, showed a half-life of 6.7 days (95% CI 4.9 to 10.5 days). Duplicated experiments showed the same pattern. These results indicate that some parameters associated with the presence of *Microcystis*, especially *M. aeruginosa*, appear to be detrimental to *E. coli* survival.

### 2.2. Growth of Microcystis Cultures in the Presence of E. coli

The growth of cyanobacteria was monitored for the possible effects of *E. coli*. During the same experiment that provided the results of *E.coli* growth with and without *Microcystis* present (Figure 1), Chlorophyll *a* concentrations and cell counts of *Microcystis* cultures with and without *E. coli* were measured and are plotted in Figure 2. Figure 2a shows that chlorophyll *a* concentrations in *M. smithii* and *M. aeruginosa* both increased slowly over the first eight days and then rose rapidly through day 16. Cell counts of the two *Microcystis* species (Figure 2b) also exhibited a significant increase, except with the increase being noticed after five days. The similar trends in the chlorophyll *a* and cell count results for cultures with and without *E. coli* indicate that the presence of *E. coli* had little impact on *Microcystis* growth.

### 2.3. pH and Microcystin Levels in Microcystis Cultures in Relation to E. coli Survival

Laboratory measurements investigated possible links between *E. coli* survival, pH, and microcystin levels associated with the growth of *Microcystis*. pH is an important variable affecting aquatic environments and habitat dynamics [4,20]. To investigate the possible influence of pH, it was monitored throughout the growth period of *M. aeruginosa* and *M. smithii* cultures and showed gradual increases from 8.25 ± 0.08 to 9.59 ± 0.04 and 9.88 ± 0.05 (mean ± SD), respectively. The pH of the control culture with only *E. coli* remained within 0.02 units of the initial pH of 8.25 over 28 days. *E. coli* was still alive in *M. smithii* when pH reached 8.52 ± 0.04 on Day 8, but already absent in *M. aeruginosa* when pH reached 8.34 ± 0.03 on Day 5. 

To determine the potential role of microcystin in these *E. coli* survival studies, microcystin levels were measured throughout the culture period. The results are shown in Figure 3. Combined with the growth variables in Figure 1, it is obvious that microcystin concentrations in both species increase with the growth of the cyanobacteria. Meanwhile, *M. aeruginosa* constantly produced more microcystin (from 1.12 to 26.7 ppb) than *M. smithii* (from 0.60 to 15.0 ppb) at comparable chlorophyll *a*/cell count levels, with a statistically significant difference at day 23 (*p* = 0.001).

To assess the possible effect of direct cellular competition with *Microcystis* on the decline of *E. coli*, MA culture medium was prepared with different concentrations of microcystins using sequential dilutions of filtered *M. aeruginosa* culture containing 20.0 μg/L microcystin. The same level of *E. coli* consortium (40,000 CFU/100 mL) was then inoculated into 3 L of each culture medium and monitored for *E. coli* survival. A plot of *E. coli* levels versus time (Figure 4) showed an initial increase at low microcystin concentrations with a decrease beginning on day 2, while the *E. coli* levels at 5.0 μg/L decreased from inoculation. This may indicate that 2.0 μg/L microcystin was lower than the concentrations for the earlier experiments (Figure 1 plots with Microcystis present) and that the dynamics of microcystin toxicity to *E. coli* are more complicated. The results show that the survival time of *E. coli* decreased from 20 to 5 days as microcystin concentration increased from 0.5 to 5.0 μg/L (Table 1). Using the data starting at day 2 when a pseudo-first-order decrease in *E. coli* was apparent, the half-life (*t*_1/2_) of *E. coli* was calculated (Table 1) and followed a corresponding pattern to survival time at increasing microcystin levels. Statistical analysis showed that *E. coli* survival was significantly reduced at all microcystin concentrations compared with the control, and the reduction at 2.0 μg/L was greater than at 0.5 and 1.0 μg/L (*p* < 0.05). 

### 2.4. Survival of E. coli and Total Coliforms from Lake Water in Microcystis Cultures

Lake water containing a natural consortium of *E. coli* and total coliform bacteria was collected at a location known to receive combined-sewage outflows from Lake Ontario at Kingston, Ontario, Canada. The lake water was determined to have *E. coli* and total coliform levels of 6450 and 43,520 CFU/100 mL, respectively. Lake water was inoculated into the *M. smithii* and *M. aeruginosa* cultures, along with an MA medium control sample, to explore the survival of natural bacteria strains under the same conditions studied for the laboratory strains. As seen in Figure 5, the lake water bacterial *E. coli* and coliform strains showed similar survival patterns to the *E. coli* lab strains. They survived longer in *M. smithii* (2–3 days) than in *M. aeruginosa* (1 day), while both bacteria levels remained high for over 14 days in the control medium. The regression analysis of log_e_-transformed *E. coli* levels in *M. smithii* culture was not statistically significant (*p* = 0.089), so the half-life could only be estimated as 0.38 days. The half-life value for *E. coli* in *M. aeruginosa* culture was 0.33 days (95% CI 0.31 to 0.35 days), consistent with the faster decay compared with the *M. smithii* culture. Starting from day 1, the control culture showed a half-life of 6.5 days (95% CI 3.9 to 20.0 days). Total coliforms also had a longer half-life in *M. smithii* (0.93 days, 95% CI 0.71 to 1.35 days) than in *M. aeruginosa* (0.36 days, 95% CI 0.28 to 0.51 days). Starting from day 1, the control culture showed a total coliform half-life of 6.2 days (95% CI 3.7 to 18.9 days). Meanwhile, a non-toxin-producing green algae, *Chlorella* (identified by microscopy), was observed to grow in the control sample without any effect on the survival of *E. coli* and total coliforms (Figure 5). It was noted that the starting concentrations (day 0) of *E. coli* and total coliforms were lower in the *Microcystis* cultures than in the control, unlike the case for inoculating lab strains of *E. coli* as shown in Figure 1. The exact explanation of this difference is not known, but there may be a more immediate effect of the Microcystic culture on the survival of natural *E. coli* and total coliform strains in lake water such that it was seen immediately after the short time (several minutes) between mixing the lake water into the culture and withdrawing the samples for enumeration. This should be studied more extensively in the future. Nonetheless, we believe a general comparison of bacteria survival and measured half-life values following lake water inoculations to the laboratory isolate inoculations can still be made. 

## 3. Discussion

The cyanobacteria-*E. coli* dynamic investigation shows that *E. coli* survival was severely affected (shortened) by the presence of *Microcystis*, especially by *M. aeruginosa*. However, this could not be attributed to a large degree to changes in pH. During pH monitoring throughout cyanobacteria growth and *E. coli* decline, it was observed that pH in both *M. aeruginosa* and *M. smithii* increased as expected during photosynthetic activity consuming CO_2_ [21,22,23].

Although the increase in pH exceeded the optimal range for *E. coli* growth, it was still in their reported bearable range. Even if elevated pH could combine with other adverse factors to reduce the survival of *E. coli* in the two *Microcystis* cultures, it cannot explain the significant difference between the effects of the two *Microcystis* species, where the effect is less in the higher pH of *M. smithii*. Nevertheless, high pH might serve as an auxiliary element in the decline of *E. coli*. On the other hand, *M. aeruginosa* constantly produced more microcystin (from 1.12 to 26.7 ppb) than *M. smithii* (from 0.60 to 15.0 ppb) at comparable chlorophyll *a*/cell count levels. This is the only parameter studied that differed significantly between the *M. aeruginosa* and *M. smithii* cultures, which makes microcystins the most likely candidate for explaining the difference in survival of *E. coli* between the cultures.

The major causes of the decline in *E. coli* are not likely to be direct contact with the cyanobacteria or competition for resources within the medium given the results in the filtered cell-free treatments with microcystin (Figure 4). Compared to the cultures with cyanobacteria growing at the same time (Figure 1), *E. coli* generally had longer survival times (longer half-life) in the filtered samples at similar microcystin levels. These results are consistent with the previous suggestion that microcystin production might be the main cause of reduced *E. coli* survival, though other factors such as high pH values and the presence of *Microcystis* cells cannot be entirely ruled out. 

This study indicates that the major cause of the decline in *E. coli* was linked to the presence of microcystins, which is consistent with these being the most significant toxins produced by *Microcystis* species. This is of particular interest because the levels of microcystins in this study were comparable to those found in the local waters of Lake Ontario during HAB events [24]. Given the high toxicity to a variety of organisms, microcystins have long been assumed to serve a defensive function against predators or competitors [25]. When a natural consortium of *E. coli* and total coliform bacteria from Lake Ontario was inoculated into the *M. smithii* and *M. aeruginosa* cultures to explore the survival of natural bacteria strains under the same conditions studied for the laboratory strains, the results were similar to that of the latter. In addition, when *Chlorella* was present in the absence of the *Microcystis* species in lake water, there was no deleterious effect on *E. coli*, further supporting the notion that toxin from cyanobacteria played an important role in reducing survival of the tested FIB. It is known that *Microcystis* can alter native bacterial assemblages in lakes [7], that microcystins modify bacterial communities in sediments [8] and affect bacterial growth in freshwater reservoirs [6].

More specifically, Ostensvik and colleagues [26] showed inhibition of bacteria by extracts from five different cyanobacteria, including *Microcystis aeruginosa*, that may contain microcystins, but they did not monitor the fecal indicator bacteria, *E. coli* or coliforms. Another study tested *E. coli* from sewage [27], however, it did not reveal any possible effects of cyanobacteria because the bacteria died under all investigated conditions. 

In contrast to our findings, two previous investigations did not see growth inhibition of *E. coli* challenged with pure microcystin-RR [28,29]. These investigations, however, used laboratory *E. coli* strains growing in rich nutrient broth. Such a rich medium might be expected to provide an abundance of resources with which the bacteria could cope with toxins. By comparison, the present study used environmental *E. coli* strains and coliforms found in lake water grown in a minimal medium with microcystin produced by *Microcystis*. The latter better emulates more natural conditions of real aquatic ecosystems and would be expected to provide more credible results.

Halac and colleagues [30] have reported a negative correlation suggesting a similar problem with fecal indicators to the present findings, however, they did not directly measure microcystin and so only infer the effect of toxins based on cyanobacteria levels.

Our results indicate that *E. coli* should be added to the list of bacteria affected by toxic *Microcystis*. Although native consortia of *E. coli* were eliminated by *Microcystis* in laboratory microcosms, further research is needed to determine if similar dynamic relationships are expressed in water systems containing natural populations of *Microcystis*.

The present findings point toward the possibility that *E. coli* abundance might not be a reliable indicator of faecal contamination in waters also containing toxin-producing cyanobacteria. The interpretation of *E. coli* test results should take into account the possible influence of co-existing cyanobacteria, and when tests show *E. coli* negative results, supplementary tests might be needed before concluding the absence of faecal contamination. 

On the other hand, these results could indicate a new opportunity for the use of *E. coli* and coliforms for *Microcystis* monitoring. If a water system normally has *E. coli* or coliform bacteria present, the sudden reduction or loss of these bacterial populations could indicate the onset of *Microcystis* growth as in a HAB. Therefore, the relationship between the disappearance of *E. coli* and the level of cyanobacterial toxins could be of great use for water quality management that pertains to FIB screening and HAB prediction and control. Work in this paper is an initial step in exploring the dynamics of cyanobacteria-coliform bacteria populations in laboratory cultures. Apart from microcystins, of which numerous variants are known, cyanobacteria are capable of synthesizing a whole set of secondary metabolites [31] that could be expelled into the medium. As far as is known, *M. aeruginosa* contains mainly microcystins, only occasionally accompanied by aeruginosins. Nevertheless, the impact of these other compounds on bacterial growth should be investigated.

## 4. Conclusions

The fate and persistence of an ideal faecal indicator in the natural environment should be similar to the target faecal pathogens indicated. Results in this paper indicate that this similarity may not be true if the presence of *Microcystis* affects the detection of *E. coli* while not affecting the survival of other pathogens such as viruses and protozoa. Because many protozoa are known to grow or survive in the presence of HABs [32], they may present alternative choices for faecal indicators, rather than *E. coli* and coliforms, in water bodies susceptible to blooms. Low levels of *E. coli* in water samples impacted by HABs may result in a false assessment of faecal contamination risk, even if the presence of HABs may eventually be noticed and trigger a separate warning about water quality.

## 5. Materials and Methods

### 5.1. Bacterial Cultures

Reference *E. coli* strains, ATCC 25922, 8739, 51813, and 11775, were purchased in lyophilized form (Med-Ox Diagnostics, Ottawa, ON, Canada) and reconstituted and cultured in Tryptic Soy Broth. In addition, lake water was collected from Lake Ontario near Kingston, ON, as a source of naturally occurring strains of *E. coli* and total coliforms. Determination of *E. coli* and coliforms used the TECTA™ B16 microbiology system with TECTA™ CCA test cartridges (TECTA-PDS, now IDEXX-Tecta, Kingston, ON, Canada) described below in Section 5.4, and with additional analysis using Colilert-18 with Quanti-Tray (IDEXX Laboratories, Inc., Westbrook, ME, USA).

### 5.2. Cultures of Cyanobacteria

Two *Microcystis* species, *M. smithii* and *M. aeruginosa*, were isolated from Leonard’s Fish Farm, a collection of small baitfish ponds in Hartington, ON, Canada and cultured in MA medium [19] to establish axenic stock cultures. The chemicals used were analytical grade or the highest purity available from Thermo Fisher Scientific (Ottawa, ON, Canada). Treatments were performed in 4 L Erlenmeyer flasks containing 3 L of autoclaved MA medium. All cultures were incubated with continuous aeration using filtered air (Cytiva Whatman™ ReZist Air Venting Filter, 50 mm, PTFE, 0.2μm; Thermo Fisher) at 22 ± 1°C, illuminated by four 32-watt T8 48-inch cool white fluorescent tubes at 220 W·.m^−2^ on a 16:8 light:dark cycle.

### 5.3. Addition of E. coli Consortium to Cyanobacterial Culture Media and Control Medium

Treatments were inoculated with *M. smithii* (flasks 1 and 2), *M. aeruginosa* (flasks 3 and 4) or no cyanobacteria (flask 5) and incubated for 3 days. Then an *E. coli* consortium comprised of four lab stains, ATCC 25922, 8739, 51813, and 11775 in a 1:1:1:1 mixture, was added to flasks 1, 3, and 5 at approximately 40,000 CFUs (colony forming units, 10,000 CFUs for each strain) per 100 mL. These three flasks were sampled and monitored for *E. coli* levels daily after the inoculation. Flasks 2 and 4 served as the no-*E. coli* control groups. Concurrently, flasks 1–4 were sampled to monitor cyanobacterial growth. The data presented is from one of three independent experiments. The *E. coli* consortium employed in these experiments more closely represents the diversity of strains that occur in lake water compared to using single strains.

### 5.4. E. coli and Total Coliform Detection Using the TECTA Microbiology System

The TECTA B16 system and test cartridge provides a culture-based method for *E. coli* and coliform testing with automated instrumental detection of fluorescent products of selective enzyme-substrate reactions [33,34]. It features a polymer-based optical sensor that extracts the fluorescent products from the broth culture to enable detection even in the presence of color and turbidity from other organisms. Bacteria levels in the initial sample are estimated by noting the time-to-detection (TTD), which varies inversely with the logarithm (base ten) of the bacteria level. The TECTA B16 offers an additional advantage of a wide dynamic range so that samples with unknown bacteria levels can be analyzed without running multiple dilutions. To calibrate the TECTA response in the testing matrix, culture media spiked with known amounts of *E. coli *(confirmed with separate tests using Colilert-18) were prepared to produce five test levels ranging from 490 to 4900 CFU/100 mL. Three 100 mL calibration samples of each level were tested, and the average TTD for each set was plotted against the log of the *E coli* counts. The linear regression equation of this plot (R² = 0.97) was used for *E. coli* quantitation in subsequent experiments. For most experiments, a 10 mL sample was added to a test cartridge containing 90 mL sterile water (10-fold dilution) to reduce consumption of medium while still being sensitive enough for tracking bacteria populations. Tests were run in triplicate. A separate positive control with *E. coli* inoculated directly in the test cartridge showed that the presence of cyanobacteria did not affect the TTD value for 10 mL samples even at the 1000 cells/μL level.

### 5.5. Chlorophyll a Measurement and Cell Counts to Monitor Algal Growth

Chlorophyll *a* is a major microphyte pigment and was used as an indicator of microphyte biomass. The measurement procedure was a standard method using visible wavelength spectrometry [35]. Briefly, a 10 to 50 mL volume of sample was filtered, depending on the estimated chlorophyll *a* content, through Whatman 47 mm GF/F glass microfiber filters (VWR Canlab, Mississauga, ON, Canada) of 0.8 µm pore size, applying vacuum filtration until the filter was dry. The filter was folded into quarters, wrapped in aluminum foil, and stored frozen (−20 °C) in ziplock bags for at least 24 h. Frozen samples could be held for up to 28 days from the time of filtration until extraction. For chlorophyll *a* extraction, a freeze/thaw cycle (20 min/5 min) was repeated three times to break the cell membranes. Sample filters were then removed from the foil wrap, placed in a 15 mL centrifuge tube, and steeped in 10 mL 90% acetone to extract chlorophyll from the microphyte cells. The centrifuge tubes were vortexed vigorously to mix well and then stored at 4 °C for 8–10 h, to allow for complete extraction. The extract was clarified by centrifuging for 15 min at 2200× *g*. 1.0 mL samples of the clear supernatant were carefully transferred to 1 cm pathlength quartz cuvettes and absorbances at 750, 664, 647, and 630 nm were measured using a SpectraMax Plus 384 spectrophotometer (Molecular Devices, Sunnyvale, CA, USA). The chlorophyll *a* concentration was calculated using Jeffrey and Humphrey’s Trichromatic Equation (18) [35]:$$Chlorophyll\;a\;\left(\frac{\mu g}{L}\right)=\frac{\left[11.85\left(A_{664}-A_{750}\right)-1.54\left(A_{647}-A_{750}\right)-0.08\left(A_{630}-A_{750}\right)\right]V_{1}}{V_{2}L}$$
where *A_n_* = absorbance at wavelength *n*, *V*_1_ = the volume of 90% acetone used for extraction (mL), *V*_2_ = the volume of water filtered (L), and *L* = the cell path length (cm). 

The *Microcystis* species consisted of single cells as cultured and cell counts were determined using a hemocytometer (Qiujing Biochemical Reagent and Instrument Co., Ltd., Shanghai, China) and a Labophot microscope (Nikon, Japan) at 400× magnification with camera attachment.

### 5.6. Microcystin Determination

Microcystins are cyclic peptides with seven amino acids and are the most common toxins produced by cyanobacteria. They are named for the various amino acids in their structures, with Microcystin-LR being the most common and also the most toxic. Microcystins were determined by Enzyme-Linked Immunosorbent Assay (ELISA) using a QuantiPlateTM Kit for Microcystins (Envirologix, Portland, ME, USA). This is a competitive ELISA where microcystin-labelled horseradish peroxidase enzyme competes with microcystin from the sample to bind with an immobilized antibody. After a washing step, enzyme activity is determined with a colourimetric substrate, giving a detection limit of 0.2 ppb reported by the manufacturer. This assay only detects extracellular microcystins because cells were not disrupted during this analysis. Moreover, this assay does not distinguish between the microcystin toxin variants. Microcystin in the sample competed with enzyme (horseradish peroxidase)-labeled microcystin for a limited number of antibodies immobilized on the inside surface of the test wells. After washing off the non-binding material, a color development step produced a solution with absorbance that was inversely proportional to microcystin concentration. A calibration curve was produced (R^2^ = 0.99) using standard calibrator solutions, according to the manufacturer’s instructions.

### 5.7. Determination of Half-Life for Decline of Bacteria Cultures

Culture half-life (t12) for cases where there was a bacteria population decrease over time (Figure 1, Figure 4 and Figure 5) was determined by assuming a pseudo-first-order kinetic decline. The slope of a plot of log_e_(population) vs. time was determined by linear regression, and the half-life was calculated as 0.693/(−1 × slope). In samples where there was an increase in population followed by a decrease, the starting day for the regression was shifted as noted in the text. The 95% confidence interval range for the half-life was determined by calculating lower and upper limit values from the 95% confidence interval of the slope. Since the half-life value is an inverse of the slope values, the 95% CI range is not symmetrical around the mean half-life value, so the two range limits are given rather than plus/minus a single value.

## Figures and Tables

**Figure 1 toxins-15-00628-f001:**
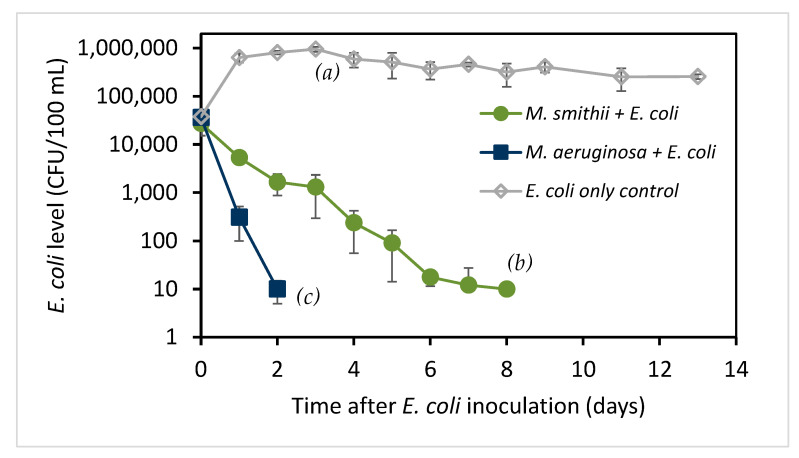
*E. coli* levels (CFU/100 mL) in *Microcystis* cultures and the control medium after inoculation with *E. coli* (mean ± SD; n = 4). Statistical testing on day 2 shows that both *Microcystis* cultures are significantly different from the control (*p* = 3 × 10^−7^ for both) and that *M. aeruginosa* culture levels are different from *M. smithii* (*p* = 0.006), as indicated by letters *a*, *b*, *c*.

**Figure 2 toxins-15-00628-f002:**
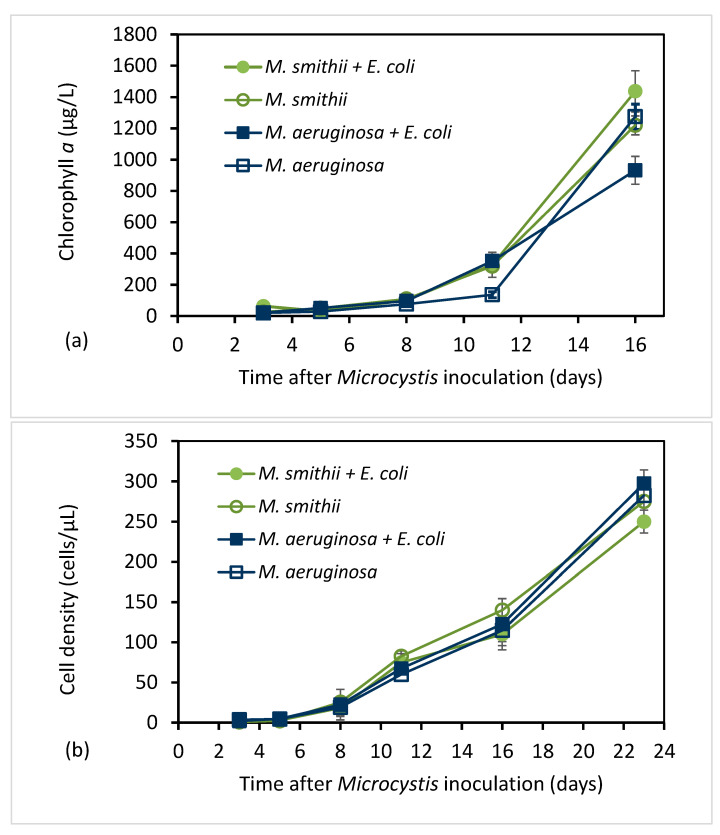
Chlorophyll *a* concentration (µg/L) (**a**) and cell density (cells/µL) (**b**) in *Microcystis* cultures with and without *E. coli* recorded on various days after inoculation of the culture (mean ± SD; n = 3).

**Figure 3 toxins-15-00628-f003:**
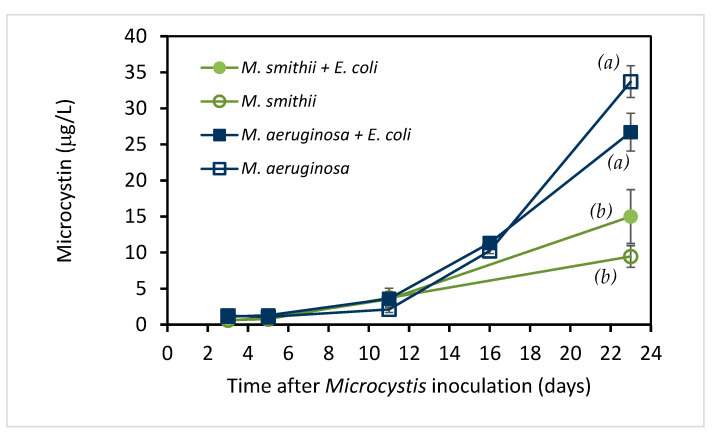
Microcystin concentration (µg/L) in two microcystin producing *Microcystis* cultures with and without *E. coli* recorded on various days after inoculation of the culture (means ± SD; n = 2). By day 23, the microcystin levels with and without *E. coli* present were not significantly different, but the microcystin levels in *M. aeruginosa* cultures were significantly different from *M. smithii* cultures (*p* = 0.001), as indicated by letters *a*, *b*.

**Figure 4 toxins-15-00628-f004:**
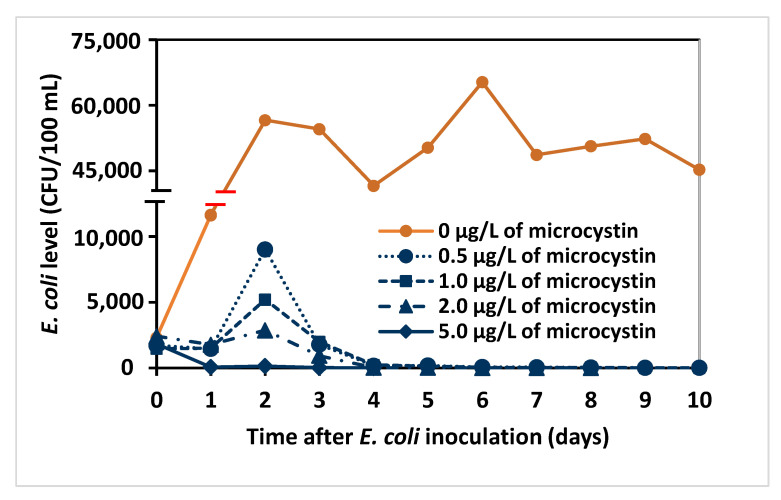
*E. coli* counts (CFU/100 mL) in MA cultures containing different levels of microcystin adjusted with cell-free filtrate of *M. aeruginosa* culture containing 20.0 µg/L microcystin.

**Figure 5 toxins-15-00628-f005:**
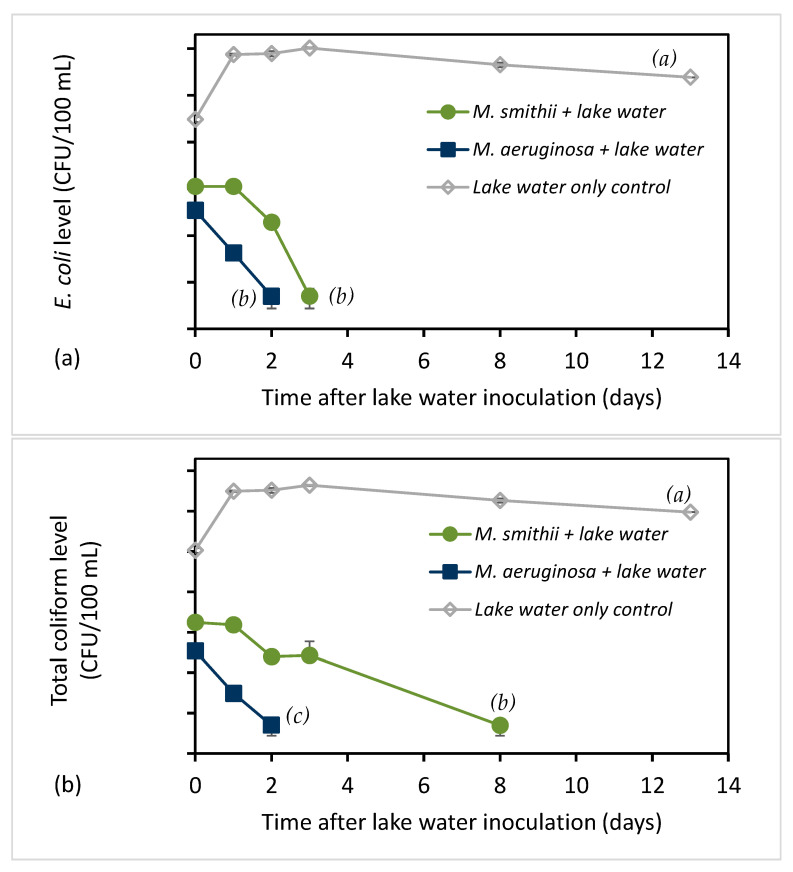
Survival of lake strain *E. coli* shown in level (CFU/100 mL) over time (days) (**a**) and total coliforms level (CFU/100 mL) (**b**) in two *Microcystis* cultures and control medium recorded on various days after inoculation of the culture (means ± SD; n = 2). Letters indicate curves that are statistically significantly different from each other (*p *< 0.05).

**Table 1 toxins-15-00628-t001:** *E. coli* survival time (last day with bacteria detected) and half-life (*t*_1/2_) in cultures with different levels of microcystin. Half-life (*t*_1/2_) and 95% CI ranges were determined by linear regression of log_e_-transformed data from Figure 4, starting at day 2 (see Section 5).

Microcystin (μg/L)	0	0.5	1.0	2.0	5.0
**Survival time (day)**	>25	20	18	9	5
*t* _1/2_ **(95% CI range)**	>13.3 ^a^	0.97 ^b^ (0.79–1.25)	0.74 ^b^ (0.62–0.93)	0.33 ^c^ (0.23–0.58)	0.40 ^b,c^ (0.24–1.19)

a, b, c indicate results in statistically significantly different groups (*p* < 0.05).

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
