# Peer review of "The Effect of Microcystis on the Monitoring of Faecal Indicator Bacteria"

_toxins, 2023, doi:10.3390/toxins15110628_

Round 1

Reviewer 1 Report

Big picture: The authors are attempting to prove a narrative showing that microcystins and/or microcystis can significantly impact E coli growth in media containing both. These findings reported are interesting and show interesting differences from literature precedent, but at present there are enough issues teasing out experimental details and controls that it is difficult to know how much weight to give the conclusions. A few additional control experiments (if not at least additional comparisons with prior experiments to give context) and more clear experimental details would be helpful for better proving these claims. Finally, the prior work in this area in the literature should be better discussed with regard to the conclusions presented.

Specific concerns

  1. Results section 2.1. A control experiment where a non-microcystis organism is incubated then exposed to the same E. coli. addition would further help elucidate whether outcompetition or the direct impact of microcystis, microcystins, or some other specific factor impacting the growth of E. coli. As it stands now the data presented do not show a compelling case for why it can't just be an established colony outcompeting a new arrival (such as there being no nutrient availability).

  1. Related, it would likely help clarify this question if the concentrations of microcystis and E. coli were showed side-by-side for the same experiments. For example, in lines 72-74, E. coli are added to a microcystis culture, but the details of that culture's cell density, age, whether it had reached maximum growth, etc. are not clear. If this is the experiment described in lines 245-248 (which needs to be specified somehow), then Figure 1 is particularly surprising because  Figure 2 suggests that after 3 days of incubation cell densities of microcystis were roughly 3 orders of magnitude lower than where they were after 21 days of incubation, while Figure 1 shows the E. coli reaching saturation after only a day and a half in the control culture. If these are the same temporal axis, then showing the slow rise in microcystis concomittant with the drop in E. coli would go a long way toward proving this piont.

  1. Further, this reviewer is not a microbiologist, but the use of cells/mL for microcystis and CFUs/mL for E. coli is making it difficult for a direct comparison of the cell densities in these cultures. Would it be possible to use a consistent unit, or if not, at least provide context for the relative concentrations indicated in the experiments in this paper?

  1. Results section 2.3. It would be interesting to examine the use of purified standards of microcystins (i.e. not culture filtrate) to isolate the potential impact of this toxin from any other soluble material in the diluted culture media added in section 2.3. As the authors themselves state there may be a variety of other toxins or compounds produced which could have an impact on cyanobacterial growth (lines 210-214). The authors may not want to perform a new experiment, but the use of added culture media adds more variables than simply the toxins. An examination of literature work related to microcystin interactions with E. coli would be helpful if more experimental work is not done. 

  1. Figure 4. Would it be possible to include the data for the control E. coli level here as well? Table 1 implies a >25 day t1/2 for the control sample, which is strikingly different from any of the cultures containing any microcystins, even at a relatively low concentration of 0.5 ug/L. Related, is the 1 day data available to include in Figure 4? Table 1 further suggests t1/2 of 0.34 and 0.32 days for the high concentrations of MCs, which suggests a dramatic drop would be expected by day 2. Having day 1 would thus be very important for showing how these values were determined, as multiple orders of magnitude of decrease have clearly occurred before the first data points in Figure 1.

  1. Lines 186-187: The authors assert that Chorella was 'substituted' for the Microcystis in the culture, as a control. This seems to be misrepresenting what occurred: as described in lines 133-147, it is not that Chorella were present as an established colony to which the E. coli were added, but rather that they were seen by microscopy in the lakewater samples.

  1. Discussion: As mentioned above, there could easily be more details provided regarding past studies looking at the impact of microcystins on E coli and how the findings here relate. As it stands now the discussion claims (line 190) that no previous work has been done in this area, which is inaccurate. Indeed, there is some controversy regarding the impact of microcystins/microcystis on E. coli, whether minimal or otherwise, and this should be discussed here.

Author Response

See attached file for Reviewer 1.

Reviewer 2 Report

The authors investigated the influence of Microcystis cyanobacteria on the survival of E. coli. The manuscript is interesting and well-written, but can be further improved after addressing the following comments:

1.      The aim and objectives of the study could be clearly spelt out in the abstract.

2.      The authors could briefly explore why Microcystis cyanobacteria has such an effect on E. coli.

3.      In view of the results, could an alternative FIB could be made?

Author Response

See attached file for response to Reviewer 2.

Reviewer 3 Report

Journal: Toxins (ISSN 2072-6651)

Manuscript ID: toxins-2533640

Type: Article

Title: The effect of cyanobacteria of the genus Microcystis on the monitoring of faecal indicator bacteria

Section: Marine and Freshwater Toxins

Special Issue: Harmful Algal Blooms and Microalgae from Ecology to Control Strategies

This manuscript aimed to study the effect of Microcystis on the monitoring of faecal indicator bacteria. However, the presentation and writing are not good. There is lack of statistical analysis or repeated experiment, thus the data is not robust. I have the following comments and suggestions for the authors to improve the quality of the manuscript.

1. Title

“The effect of cyanobacteria of the genus Microcystis on the monitoring of faecal indicator bacteria.”

Please change to “The effect of Microcystis on the monitoring of faecal indicator bacteria”.

2. Abstract

Lines 5-7

“The survival of Escherichia coli (E. coli) bacteria, the most common faecal indicator bacteria (FIB), may be significantly affected by Microcystis cyanobacteria such as would be present during a harmful algal bloom (HAB).”

Please delete “such as would be”.

3. Section “1. Introduction”

Lines 22-25

“Water bodies in North America and around the world have experienced increased algal blooms, including those harmful algal blooms (HABs) caused by cyanobacteria as a result of enhanced human- and climate-driven environmental changes. HABs affect the ecological integrity of the affected water by disrupting or altering the biological communities.”

Please cite the following paper.

Challenges of using blooms of Microcystis spp. in animal feeds: A comprehensive review of nutritional, toxicological and microbial health evaluation. https://doi.org/10.1016/j.scitotenv.2020.142319

4. Lines 22-24

“Water bodies in North America and around the world have experienced increased algal blooms, including those harmful algal blooms (HABs) caused by cyanobacteria as a result of enhanced human- and climate-driven environmental changes.”

Please delete “those”.

5. Section “5. Materials and Methods”

Lines 245-246

“smithii in flask #1 and #2 and M. aeruginosa in flask #3 and #4 were incubated for three 245

days.”

6. Section “5.3. Addition of E. coli consortium to cyanobacterial culture media and control medium M.”

Did you perform repeated experiments? Please present number of repeated experiments, n.

7. Section “5.3. Addition of E. coli consortium to cyanobacterial culture media and control medium M.”

What is the density of Microcystis when the experiments begin? Please present the information in the revised manuscript.

8. Please draw a figure to show the design of experiments, flasks, groups, sampling time and detection of the parameters, number of replicates for experimental groups and detection of each parameter.

9. Section “5.6. Microcystin determination”

What is the detection of limit for concentrations of microcystins? Please present the information in the revised manuscript.

10. Please insert a sub-section about statistical analysis in the section “5. Materials and Methods”

11. Four strains of E. coli (ATCC 25922, 8739, 51813, and 11775) were used for experiments, but in the results section, there is no information about strains? Did the Microcystis cause the same effects on different strains?

12. Fig. 1, 2, 3, 5

Please present results of statistical analysis. Please use different letters to show significant differences among groups. For example, "a" and "b" or "bc" or "c" have significant differences, but "b" and "ab" or "bc" have no significant differences. It is meaningless if there are no results of statistical analysis.

13. Fig. 2a

Did you perform repeated experiments? Please present data of standard deviation (SD).

14. Lines 100-105

“To investigate the possible influence of pH, it was monitored throughout the growth period of M. aeruginosa and M. smithii cultures and showed gradual increases from 8.2 to 9.6 and 9.9, respectively. The pH of the control culture with only E. coli remained close to the initial pH of 8.2 over 28 days. E. coli was still alive in M. smithii when pH reached 8.8 on Day 9, but already 104

absent in M. aeruginosa when pH reached 8.4 on Day 6.”

Please present data of pH in a figure or a table. Please present data of standard deviation (SD) and results of statistical analysis.

15. Table 1

“E. coli survival time (last day with bacteria detected) and half-life (t1/2) in cultures with different levels of microcystin.”

What do you mean by half-life (t1/2)? Please insert more details in the revised manuscript.

16. Fig. 4

Did you perform repeated experiments? Please present data of standard deviation (SD).

17. Fig. 5

On day 0, data of E. coli and total coliforms level in two Microcystis cultures and control medium were very different. The conditions should be the same at the start of experiments, on day 0. If the conditions of E. coli and total coliforms are different, you can not say the different results on days 2 and after were caused by Microcystis.

major edits

Author Response

See attached file for response to Reviewer 3.

Round 2

Reviewer 1 Report

The authors have made a number of positive changes to the paper in response to the prior comments. In general these have strengthened the impact of the results, and better provided context with regard to prior results in the literature. At this point the only significant comment I have relates to a change made to Figure 4/Table 1, which were updated in response to the prior comments. 

Lines 140-150: The authors have added the results for T=0 to 2 d to Figure 4, and it's interesting to see an initial increase in E. coli in the lower concentration toxin samples from the starting level before the eventual drop-off. The overall indication remains that the toxins are deleterious to the growth of the E. coli, but the effect seems to be much more dependent on toxin concentration, and possibly even something with a delayed onset. I can see why the initial points were 'pruned', but I think that the picture presented now is much more interesting, while still supporting the conclusions posted.

I think an update of the text here to address this section of the figure should be done. As it is now the T1/2 calculation in Table 1 was updated to include where it began (at 2 days) without discussion in the text as to why this interval was chosen. Specific topics to consider including here or in the discussion would be potential comparison with the Figure 1 competition experiment, and the initial growth phase in the lower toxin concentration samples. 

Author Response

Response: The text in this section has been modified to include these points by the reviewer.  The modified text starting at line 146 now reads:

“A plot of E. coli levels versus time (Figure 4) showed an initial increase at low microcystin concentrations with a decrease beginning on day 2, while the E. coli levels at 5.0 μg/L decreased from inoculation. This may indicate that 2.0 μg/L microcystin was lower than the concentrations for the earlier experiments (Fig. 1 plots with Microcystis present) and that the dynamics of microcystin toxicity to E. coli are more complicated. The results show that the survival time of E. coli decreased from 20 to 5 days as microcystin concentration increased from 0.5 to 5.0 μg/L (Table 1). Using the data starting at day 2 when a pseudo-first order decrease of E. coli was apparent, the half-life (t½) of E. coli was calculated (Table 1) and followed a corresponding pattern to survival time at increasing microcystin levels.”

Reviewer 3 Report

Journal: Toxins (ISSN 2072-6651)

Manuscript ID: toxins-2533640-peer-review-v2

Type: Article

Title: The effect of Microcystis on the monitoring of faecal indicator bacteria

Section: Marine and Freshwater Toxins

Special Issue: Harmful Algal Blooms and Microalgae from Ecology to Control Strategies

This manuscript aimed to study the effect of Microcystis on the monitoring of faecal indicator bacteria.

The revised manuscript improved to some degree. The writing and presentation are not good yet. Also, statistical analyses were not performed. I have the following comments and suggestions for the authors to improve the quality of the manuscript.

1. Please draw a figure to show the design of experiments, flasks, groups, sampling time and detection of the parameters, number of replicates for experimental groups and detection of each parameter.

I have made this comment last time. But the reviewers did not insert the figure. The figure can help readers better understand your paper.

2. Lines 363-366

“Microcystins were determined by Enzyme-Linked Immunosorbent Assay (ELISA) using a QuantiPlateTM Kit for Microcystins (Envirologix, Portland, ME) with a detection limit of 0.2 ppb.”

Which kind of ELISA did you use? There are several ELISA methods for microcystins. More details are needed.

3. Fig. 1, 2, 3, 4, 5

Please present results of statistical analysis. Please use different letters to show significant differences among groups. For example, "a" and "b" or "bc" or "c" have significant differences, but "b" and "ab" or "bc" have no significant differences. It is meaningless if there are no results of statistical analysis.

I have made this comment last time. But the reviewers did not insert the results of statistical analysis in the figures.

Please refer to the following paper.

Influence of Iron on Physiological Parameters and Intracellular Microcystin in Microcystis Panniformis Strain Isolated from a Reservoir in the Amazon. https://doi.org/10.1007/s00284-021-02499-5

major edits

Author Response

The revised manuscript improved to some degree. The writing and presentation are not good yet. Also, statistical analyses were not performed. I have the following comments and suggestions for the authors to improve the quality of the manuscript. 

Response: Perhaps the reviewer did not see the statistical analysis results that were added to all relevant figure captions and in the text.  We believe all data comparisons are now supported by appropriate statistical analyses.

  1. Please draw a figure to show the design of experiments, flasks, groups, sampling time and detection of the parameters, number of replicates for experimental groups and detection of each parameter.

I have made this comment last time. But the reviewers did not insert the figure. The figure can help readers better understand your paper.

Response: Perhaps the reviewer did not note our response to this point in the previous submission.  We do not believe a simple figure with design of experiments can be prepared for this manuscript given the range of different procedures described.  We did note several procedure-related clarifications that the reviewer posed and therefore added detailed clarifications to the text.  We believe this accomplishes the same goals as a proposed design of experiments figure.  The new text starting at line 310 now reads:

“Treatments were inoculated with M. smithii (flasks 1 and 2), M. aeruginosa (flasks 3 and 4) or no cyanobacteria (flask 5)  Then an E. coli consortium comprised of four lab stains, ATCC 25922, 8739, 51813, and 11775 in a 1:1:1:1 mixture, was added to flasks 1, 3, and 5 at approximately 40,000 CFUs (colony forming units, 10,000 CFUs for each strain) per 100 mL. These three flasks were sampled and monitored for E. coli level daily after the inoculation. Flasks 2 and 4 served as the no E. coli control groups. Concurrently, flasks 1-4 were sampled to monitor cyanobacterial growth. Data presented is from one of three independent experiments.  The E. coli consortium employed in these experiments more closely represents the diversity of strains that occurs in lake water compared to using single strains.”   

  1. Lines 363-366

“Microcystins were determined by Enzyme-Linked Immunosorbent Assay (ELISA) using a QuantiPlateTM Kit for Microcystins (Envirologix, Portland, ME) with a detection limit of 0.2 ppb.”

Which kind of ELISA did you use? There are several ELISA methods for microcystins. More details are needed.

Response: the text starting at line 370 now reads: Microcystins were determined by Enzyme-Linked Immunosorbent Assay (ELISA) using a QuantiPlateTM Kit for Microcystins (Envirologix, Portland, ME). This is a competitive ELISA where microcystin-labelled horseradish peroxidase enzyme competes with microcystin from the sample to bind with an immobilized antibody. After a washing step, enzyme activity is determined with a colourimetric substrate, giving a detection limit of 0.2 ppb reported by the manufacturer.”

  1. Fig. 1, 2, 3, 4, 5

Please present results of statistical analysis. Please use different letters to show significant differences among groups. For example, "a" and "b" or "bc" or "c" have significant differences, but "b" and "ab" or "bc" have no significant differences. It is meaningless if there are no results of statistical analysis.

I have made this comment last time. But the reviewers did not insert the results of statistical analysis in the figures.

Please refer to the following paper.

Influence of Iron on Physiological Parameters and Intracellular Microcystin in Microcystis Panniformis Strain Isolated from a Reservoir in the Amazon. https://doi.org/10.1007/s00284-021-02499-5

Response: These letters are now added to Figures 1, 3 and 5.  Figure 2 had curves that were all statistically identical, and Figure 4 statistical analysis is presented in Table 1 using the suggested letters.